# A Max-Min Approach to the Worst-Case Class Separation Problem

**Mohammad Mahdi Omati**                                    *mohammad_omati@yahoo.com*
*Department of Electrical Engineering*
*Sharif University of Technology, Tehran, Iran*

**Prabhu Babu**                                              *prabhubabu@care.iitd.ac.in*
*Centre for Applied Research in Electronics (CARE)*
*Indian Institute of Technology Delhi, New Delhi-110016, India*

**Petre Stoica**                                                        *ps@it.uu.se*
*Division of Systems and Control, Department of Information Technolog*
*Uppsala University, Uppsala, Sweden 75237*

**Arash Amini**                                                     *aamini@sharif.edu*
*Department of Electrical Engineering*
*Sharif University of Technology, Tehran, Iran*

**Reviewed on OpenReview:** *https://openreview.net/forum?id=EEmwBd4tfZ*

## Abstract

In this paper, we propose a novel discriminative feature learning method based on a minorization-maximization framework for min-max (MM4MM) to address the long-standing "worst-case class separation (WCCS)" problem, which, in our design, refers to maximizing the minimum pairwise Chernoff distance between all class pairs in the low-dimensional subspace. The proposed algorithm relies on the relaxation of a semi-orthogonality constraint, which is proven to be tight at every iteration of the algorithm. To solve the worst-case class separation problem, we first introduce the vanilla version of the proposed algorithm, which requires solving a semi-definite program (SDP) at each iteration. We further simplify it to solving a quadratic program by formulating the dual of the surrogate maximization problem. We also then present reformulations of the worst-case class separation problem that enforce sparsity of the dimension-reducing matrix. The proposed algorithms are computationally efficient and are guaranteed to converge to optimal solutions. An important feature of these algorithms is that they do not require any hyperparameter tuning (except for the sparsity case, where a penalty parameter controlling sparsity must be chosen by the user). Experiments on several machine learning datasets demonstrate the effectiveness of the MM4MM approach.

## 1 Introduction

As data acquisition methods advance, modern datasets have become more complex and feature-rich, leading to potential issues such as overfitting and reduced interpretability due to redundant or irrelevant features. These challenges are particularly relevant in classification tasks, where effective feature representation is critical for optimal performance. Dimensionality reduction methods, including feature extraction Nie et al. (2021b); Wang et al. (2024); Nie et al. (2023; 2021a); Chang et al. (2016); Nie et al. (2017); Li et al. (2018a) and feature selection Gui et al. (2017); Li et al. (2017); Sheikhpour et al. (2017); Hancer et al. (2020); Li et al. (2022); Shen et al. (2021); Li et al. (2018b); Luo et al. (2018), are widely used to enhance classification accuracy by focusing on the most relevant aspects of the data.

Among the various dimensionality reduction approaches, a specific category of linear supervised methods, known as discriminant analysis (DA), has attracted considerable interest due to its focus on enhancing class separability. Multiple criteria have been proposed in the literature, each based on different definitions of separability Fisher (1936); Rao (1948); Bian & Tao (2011); Zhang & Yeung (2010); Yu et al. (2011); Su et al. (2015); Nie et al. (2021b); Wang et al. (2024). Linear Discriminant Analysis (LDA) is the most widely used technique within supervised learning, originally introduced by Fisher (1936) for binary classification and later extended to multi-class applications Rao (1948). The well-known Fisher criterion in LDA aims to identify a low-dimensional subspace that simultaneously maximizes inter-class scatter while minimizing intra-class scatter. Thanks to this approach, LDA has found extensive applications. Nonetheless, traditional LDA faces certain limitations, including the risks of over-reduction, issues with small sample size, assumptions of Gaussian-distributed data Nie et al. (2020a), Nie et al. (2020b), and sensitivity to outliers Nie et al. (2021b).

Heteroscedastic LDA (HLDA) Loog & Duin (2004) modifies traditional LDA to handle heteroscedastic cases by using the Chernoff criterion instead of the Fisher criterion. In this approach, the Chernoff distance is applied to generalize the concept of inter-class scatter. A theoretical analysis of HLDA is detailed in Peng et al.. In this class of methods, based on the definition of inter-class scatter, the Fisher criterion maximizes the arithmetic mean of all pairwise distances. This approach causes the class pair with the largest distance to dominate the projection direction, leading to an overlap of closely spaced class pairs—referred to as the "worst-case class separation" problem.

To tackle this issue, various max-min distance analysis (MMDA) methods have been introduced, based on the assumption of homoscedastic Gaussian distributions, as in Bian & Tao (2011); Zhang & Yeung (2010); Yu et al. (2011). These methods aim to maximize the minimum inter-class separability within the latent subspace. However, in real-world scenarios, classes often do not conform to the homoscedastic Gaussian assumption, meaning that differences in class means alone do not fully capture the separation between classes. Besides, existing MMDA methods attempt to increase the distance between nearest class means within the latent subspace but overlook differences in class covariances.

In Su et al. (2015), the authors introduce the heteroscedastic max-min distance analysis (HMMDA) method for dimensionality reduction, designed to leverage discriminative information derived from differences in class covariances, known as whitened HMMDA (WHMMDA). To address intra-class scatter, WHMMDA applies a preprocessing whitening step. In Wang et al. (2024), the authors propose another criterion called Max-Min Ratio Analysis (MMRA), which focuses on maximizing the minimum ratio value between inter-class and intra-class scatter to enhance the separability of overlapping pairwise classes. In both Su et al. (2015); Wang et al. (2024), to solve their specified optimization problem, the authors relax the non-convex optimization problem into a semidefinite problem, which can be solved using convex programming tools. They also propose a synthesis method to improve the precision of the solution. However, the method has a limitation: the relaxation may not be tight across all problem dimensions, potentially yielding suboptimal solutions.

In this paper, we propose an iterative algorithm for addressing the heteroscedastic max-min distance analysis problem, employing the minorization-maximization framework for min-max (MM4MM) Saini et al. (2024) to tackle the long-standing worst-case class separation (WCCS) problem. The algorithm is built upon the relaxation of a semi-orthogonality constraint, which is proved to be tight at each iteration. We first introduce a basic version of the algorithm that requires solving a semi-definite program (SDP) at each iteration. We simplify this to a quadratic program by deriving the dual of the surrogate maximization problem. Additionally, we provide reformulations that incorporate sparsity of the dimension-reducing matrix. The proposed algorithms are computationally efficient and enjoy guaranteed convergence. A notable benefit of our approach is that it requires no hyperparameter tuning, except in the sparsity case where a user-defined penalty parameter is needed. Experimental results on various machine learning datasets confirm the superior performance of the proposed approach compared to competing methods.

The paper is organized as follows: Section 2 presents the problem formulation for the worst-case class separation design and the sparse penalized problem. Section 3 details the proposed MM4MM approach. Section 4 provides numerical results, and Section 5 concludes the paper.

## 2 Problem Formulation

In this section, we formulate the worst-case class separation problem: with and without sparsity penalty. Consider the dataset $\mathbf{X} = [\mathbf{x}_1, \mathbf{x}_2, \ldots, \mathbf{x}_n] \in \mathbb{R}^{d \times n}$, where $d$ represents the dimensionality of each data point, and $n$ indicates the total number of data points, divided into $C$ classes. Let $\mathbf{m}_i$, $\boldsymbol{\Sigma}_i$, and $p_i$ represent the mean, covariance, and prior probability of class $i$, respectively, and let $\mathbf{S}_w$ denote the intra-class scatter, given by $\mathbf{S}_w = \sum_{i=1}^C p_i \boldsymbol{\Sigma}_i$. If we define the whitening transformation as $\mathbf{W}_1 = \mathbf{S}_w^{-1/2} \in \mathbb{R}^{d \times d}$, and apply it to the dataset $\mathbf{X}$, the intra-class scatter matrix becomes an identity matrix, while the covariances of different classes remain different. As a result, relying solely on differences in class means does not adequately capture class separability. Assuming each class follows a Gaussian distribution with distinct means and covariances, the Chernoff distance $d_{Cij}$ between classes $i$ and $j$ leverages discriminative information from covariance differences, enabling a more accurate description of class overlap:

$$d_{Cij} = (\hat{\mathbf{m}}_i - \hat{\mathbf{m}}_j)^T \hat{\boldsymbol{\Sigma}}_{ij}^{-1} (\hat{\mathbf{m}}_i - \hat{\mathbf{m}}_j) + \frac{1}{\alpha_{ij}(1-\alpha_{ij})} \log \frac{\left|\hat{\boldsymbol{\Sigma}}_{ij}\right|}{\left|\hat{\boldsymbol{\Sigma}}_i\right|^{\alpha_{ij}} \left|\hat{\boldsymbol{\Sigma}}_j\right|^{1-\alpha_{ij}}}, \tag{1}$$

where $\alpha_{ij} = \frac{p_i}{p_i + p_j}$, $\hat{\mathbf{m}}_i = \mathbf{W}_1^T \mathbf{m}_i$, and $\hat{\boldsymbol{\Sigma}}_i = \mathbf{W}_1^T \boldsymbol{\Sigma}_i \mathbf{W}_1$ represent the mean and variance of class $i$ in the whitened space, and $\hat{\boldsymbol{\Sigma}}_{ij} = \alpha_{ij} \hat{\boldsymbol{\Sigma}}_i + (1 - \alpha_{ij})\hat{\boldsymbol{\Sigma}}_j$. The $d_{Cij}$ can be expressed as the trace of a positive semi-definite matrix $\mathbf{S}_{Cij}$ (see, e.g., Loog & Duin (2004)):

$$\mathbf{S}_{Cij} = \hat{\boldsymbol{\Sigma}}_{ij}^{-1/2} (\hat{\mathbf{m}}_i - \hat{\mathbf{m}}_j)(\hat{\mathbf{m}}_i - \hat{\mathbf{m}}_j)^T \hat{\boldsymbol{\Sigma}}_{ij}^{-1/2} + \frac{1}{\alpha_{ij}(1-\alpha_{ij})} \left(\log \hat{\boldsymbol{\Sigma}}_{ij} - \alpha_{ij} \log \hat{\boldsymbol{\Sigma}}_i - (1-\alpha_{ij}) \log \hat{\boldsymbol{\Sigma}}_j\right). \tag{2}$$

The aim of our WCCS problem is to learn a dimension-reducing matrix $\mathbf{W} \in \mathbb{R}^{d \times d'}$, which projects the high-dimensional data $\mathbf{X}$ onto a $d'$-dimensional subspace, while maximizing the minimum pairwise Chernoff distance in the latent subspace:

$$\max_{\mathbf{W}} \min_{1 \leq i < j \leq C} \quad \mathrm{tr}\left(\mathbf{W}^T \tilde{\mathbf{S}}_{Cij} \mathbf{W}\right)$$
$$\text{s.t.} \quad \mathbf{W}^T \mathbf{W} = \mathbf{I}_{d'}, \tag{3}$$

where $\tilde{\mathbf{S}}_{Cij} = (p_i p_j)^{-1} \mathbf{S}_{Cij}$.

As an extension of WCCS problem, we also consider the problem where sparsity is imposed on $\mathbf{W}$. To achieve this, we formulate the following penalized problem:

$$\max_{\mathbf{W}} \min_{1 \leq i \leq j \leq C} \quad \mathrm{Tr}\left(\mathbf{W}^T \tilde{\mathbf{S}}_{Cij} \mathbf{W}\right) - \lambda \|\mathbf{W}\|_1$$
$$\text{s.t.} \quad \mathbf{W}^T \mathbf{W} = \mathbf{I}, \tag{4}$$

where $\|\mathbf{W}\|_1$ denotes the sum of the absolute values of the elements of $\mathbf{W}$, and $\lambda$ is a predefined penalty parameter that regulates the sparsity of $\mathbf{W}$. The formulation in (4) is inspired by prior work on sparse PCA D' aspremont et al. (2004); Zou & Xue (2018); Zou et al. (2006); Babu & Stoica (2023), where the $\ell_1$ norm of $\mathbf{W}$ is typically added as a penalty term in the objective to induce sparsity and thereby highlight the most significant elements of the estimated principal components.

In the next section, we start by reviewing the key steps of the minorization-maximization (MM) approach for solving maximization problems. We then explore how the MM approach can be adapted to address max-min problems, referred to as the MM4MM approach, which is the focus of this paper.

## 3 MM and MM4MM

### 3.1 MM framework

Consider the following constrained maximization problem:

$$\max_{\mathbf{x} \in \chi} f(\mathbf{x}), \tag{5}$$

where $\mathbf{x}$ is the variable to be optimized, $f(\mathbf{x})$ is the objective function, and $\chi$ denotes the constraint set. An MM-based algorithm addresses this problem by first creating a surrogate function $g\left(\mathbf{x} \mid \mathbf{x}^t\right)$ that serves as a lower bound for the objective function $f(\mathbf{x})$ at the current iteration point $\mathbf{x}^t$. In the second step, the algorithm maximizes this surrogate function to determine the next iterate:

$$\mathbf{x}^{t+1} \in \arg \max_{\mathbf{x} \in \chi} g\left(\mathbf{x} \mid \mathbf{x}^t\right). \tag{6}$$

The steps described above are iteratively applied until the algorithm converges to a stationary point of the problem in (7). For $g\left(\mathbf{x} \mid \mathbf{x}^t\right)$ to be considered a surrogate function, it must satisfy the following conditions:

$$g\left(\mathbf{x} \mid \mathbf{x}^t\right) \leq f(\mathbf{x}) \quad \forall \mathbf{x} \in \chi, \tag{7}$$

$$g\left(\mathbf{x}^t \mid \mathbf{x}^t\right) = f\left(\mathbf{x}^t\right). \tag{8}$$

To summarize, the main steps of the MM approach are as follows:

1) Initialize with a feasible point $\mathbf{x}^0$.

2) Construct a minorizing function $g\left(\mathbf{x} \mid \mathbf{x}^t\right)$ for $f(\mathbf{x})$ at $\mathbf{x}^t$.

3) Compute $\mathbf{x}^{t+1} \in \arg \max_{\mathbf{x} \in \chi} g\left(\mathbf{x} \mid \mathbf{x}^t\right)$.

4) If $\frac{\left|f\left(\mathbf{x}^t\right)-f\left(\mathbf{x}^{t+1}\right)\right|}{\left|f\left(\mathbf{x}^t\right)\right|} < \epsilon$, where $\epsilon$ is a predefined convergence threshold, terminate; otherwise, set $t = t + 1$ and return to Step 2.

It is easy to demonstrate that each MM step results in a monotonic increase in the objective function at every iteration, i.e.,

$$f\left(\mathbf{x}^{t+1}\right) \geq g\left(\mathbf{x}^{t+1} \mid \mathbf{x}^t\right) \geq g\left(\mathbf{x}^t \mid \mathbf{x}^t\right) = f\left(\mathbf{x}^t\right). \tag{9}$$

The first inequality and the third equality follow from (7) and (8), while the second inequality arises from (6).

## 3.2 MM4MM framework

Consider the following max-min optimization problem:

$$\max_{\mathbf{x} \in \mathcal{X}} \left\{ f(\mathbf{x}) \triangleq \min_{i=1,2,\cdots,K} f_i(\mathbf{x}) \right\}. \tag{10}$$

A surrogate function for the above max-min problem is $g\left(\mathbf{x} \mid \mathbf{x}^t\right)$, defined as follows:

$$g\left(\mathbf{x} \mid \mathbf{x}^t\right) = \min_{i=1,2,\cdots,K} g_i\left(\mathbf{x} \mid \mathbf{x}^t\right), \tag{11}$$

where each $g_i\left(\mathbf{x} \mid \mathbf{x}^t\right)$ is a tight lower bound on $f_i(\mathbf{x})$ at $\mathbf{x}^t$. The individual surrogates $g_i(\mathbf{x})$ satisfy the following conditions:

$$g_i\left(\mathbf{x}^t \mid \mathbf{x}^t\right) = f_i\left(\mathbf{x}^t\right), \tag{12}$$

$$g_i\left(\mathbf{x} \mid \mathbf{x}^t\right) \leq f_i(\mathbf{x}). \tag{13}$$

It can be readily demonstrated that the surrogate function $g\left(\mathbf{x} \mid \mathbf{x}^t\right)$, as defined in (11), meets the conditions specified in (7) and (8):

$$g_i\left(\mathbf{x} \mid \mathbf{x}^t\right) \leq f_i(\mathbf{x}) \implies \min_{i=1,2,\cdots,K} g_i\left(\mathbf{x} \mid \mathbf{x}^t\right) \leq \min_{i=1,2,\cdots,K} f_i(\mathbf{x}) \implies g\left(\mathbf{x} \mid \mathbf{x}^t\right) \leq f(\mathbf{x}), \tag{14}$$

and

$$g\left(\mathbf{x}^t \mid \mathbf{x}^t\right) = \min_{i=1,2,\cdots,K} g_i\left(\mathbf{x}^t \mid \mathbf{x}^t\right) = \min_{i=1,2,\cdots,K} f_i\left(\mathbf{x}^t\right) = f\left(\mathbf{x}^t\right). \tag{15}$$

As in the general MM framework, it can be shown here that the iterates $\{\mathbf{x}^t\}$ increase the objective function $f(\mathbf{x})$ in a monotonic manner and converge to a stationary point. For a comprehensive discussion on the MM approach—including various methods for deriving surrogate functions across different applications—refer to Sun et al. (2017); Saini et al. (2024).

### 3.3 Solving the WCCS problem

In this section, we will derive the MM4MM algorithm for the problem in (3). For the sake of convenience, we restate this optimization problem as follows:

$$\max_{\mathbf{W}} \min_{1 \leq i < j \leq C} \quad f_{ij}(\mathbf{W}) \tag{16}$$
$$\text{s.t.} \qquad \mathbf{W}^T \mathbf{W} = \mathbf{I},$$

where $f_{ij}(\mathbf{W}) \triangleq \text{Tr}\left(\mathbf{W}^T \tilde{\mathbf{S}}_{Cij} \mathbf{W}\right)$. Before proceeding with the solution to (16), we present and prove a lemma that will aid in developing the proposed algorithm.

**Lemma 1.** *The non-convex semi-orthogonality constraint $\mathbf{W}^T \mathbf{W} = \mathbf{I}$ in (16) can be relaxed to $\mathbf{W}^T \mathbf{W} \preccurlyeq \mathbf{I}$ and the global maximizer of the relaxed problem will satisfy the constraint in (16).*

*Proof.* See Appendix 6.1. $\qquad\square$

Applying Lemma 1, we reformulate the problem in (16) as the following relaxed problem:

$$\max_{\mathbf{W}} \min_{1 \leq i < j \leq C} \quad \text{Tr}\left(\mathbf{W}^T \tilde{\mathbf{S}}_{Cij} \mathbf{W}\right) \tag{17}$$
$$\text{s.t.} \qquad \mathbf{W}^T \mathbf{W} \preccurlyeq \mathbf{I}.$$

The constraint in (17) is convex since it can be rephrased as a linear matrix inequality $\begin{bmatrix} \mathbf{I}_{d'} & \mathbf{W}^T \\ \mathbf{W} & \mathbf{I}_d \end{bmatrix} \succcurlyeq 0$.

However, the maximization problem in (17) remains non-convex, as the objective function (for each $(i,j)$) is a convex quadratic in $\mathbf{W}$, and the inclusion of the min operator adds further complications. To tackle this, we employ the MM4MM approach to solve (17). Following the MM4MM steps outlined in Subsection 3.2, each convex quadratic term in (17) can be bounded from below by its tangent hyperplane at $\mathbf{W}^t$. This yields an MM surrogate for the objective in (17). Given $\mathbf{W}^t$, for any $(i,j)$, we obtain:

$$f_{ij}(\mathbf{W}) = \text{Tr}\left(\mathbf{W}^T \tilde{\mathbf{S}}_{Cij} \mathbf{W}\right) \geq \text{Tr}\left(\left(\mathbf{W}^t\right)^T \tilde{\mathbf{S}}_{Cij} \mathbf{W}^t\right) + 2\,\text{Tr}\left(\left(\mathbf{W}^t\right)^T \tilde{\mathbf{S}}_{Cij}\left(\mathbf{W} - \mathbf{W}^t\right)\right)$$
$$= 2\,\text{Tr}\left(\left(\mathbf{W}^t\right)^T \tilde{\mathbf{S}}_{Cij} \mathbf{W}\right) - \text{Tr}\left(\left(\mathbf{W}^t\right)^T \tilde{\mathbf{S}}_{Cij} \mathbf{W}^t\right) \triangleq g_{ij}(\mathbf{W}). \tag{18}$$

Then the surrogate problem is given by:

$$\max_{\mathbf{W}} \min_{1 \leq i < j \leq C} \quad g_{ij}(\mathbf{W}) \tag{19}$$
$$\text{s.t.} \qquad \begin{bmatrix} \mathbf{I}_{d'} & \mathbf{W}^T \\ \mathbf{W} & \mathbf{I}_d \end{bmatrix} \succcurlyeq 0.$$

By utilizing the expression for $g_{ij}(\mathbf{W})$ in (19), we obtain:

$$\max_{\mathbf{W}} \min_{1 \leq i < j \leq C} \quad 2\,\text{Tr}\left(\mathbf{A}_{ij}^T \mathbf{W}\right) + c_{ij} \tag{20}$$
$$\text{s.t.} \qquad \begin{bmatrix} \mathbf{I}_{d'} & \mathbf{W}^T \\ \mathbf{W} & \mathbf{I}_d \end{bmatrix} \succcurlyeq 0,$$

where

$$\mathbf{A}_{ij}^T \triangleq \left(\mathbf{W}^t\right)^T \tilde{\mathbf{S}}_{Cij}, \tag{21}$$
$$c_{ij} = -\text{Tr}\left(\left(\mathbf{W}^t\right)^T \tilde{\mathbf{S}}_{Cij} \mathbf{W}^t\right). \tag{22}$$

The problem (20) is convex and can be transformed into a semidefinite program (SDP):

$$
\begin{aligned}
\max_{\alpha, \mathbf{W}} \quad & \alpha \\
\text{s.t.} \quad & 2\operatorname{Tr}\left(\mathbf{A}_{ij}^T \mathbf{W}\right) + c_{ij} \geq \alpha, \quad 1 \leq i < j \leq C \\
& \begin{bmatrix} \mathbf{I}_{d'} & \mathbf{W}^T \\ \mathbf{W} & \mathbf{I}_d \end{bmatrix} \succcurlyeq 0,
\end{aligned}
\tag{23}
$$

which can be efficiently handled using, for example, CVX Grant & Boyd (2014). The maximizer $\mathbf{W}$ for (20) (or equivalently (23)) meets the constraint $\mathbf{W}^T \mathbf{W} = \mathbf{I}$ at every iteration of the algorithm, not solely at convergence. This notable property will be demonstrated in the following. During this process, we will also develop a computationally simpler reformulation of (23).

We start by noting that the inner minimization problem in (20) with respect to the discrete variables $i, j$ can be reformulated using auxiliary variables $\{z_{ij} \geq 0\}$ satisfying $\sum_{1 \leq i < j \leq C} z_{ij} = 1$, as shown below:

$$
\begin{aligned}
\max_{\mathbf{W}} \min_{\{z_{ij}\}} \quad & 2\operatorname{Tr}\left(\mathbf{A}^T \mathbf{W}\right) + \sum_{1 \leq i < j \leq C} z_{ij} c_{ij} \\
\text{s.t.} \quad & z_{ij} \geq 0, \quad \sum_{1 \leq i < j \leq C} z_{ij} = 1 \\
& \begin{bmatrix} \mathbf{I}_{d'} & \mathbf{W}^T \\ \mathbf{W} & \mathbf{I}_d \end{bmatrix} \succcurlyeq 0
\end{aligned}
\quad ,
\tag{24}
$$

where $\mathbf{A} \triangleq \sum_{1 \leq i < j \leq C} z_{ij} \mathbf{A}_{ij}$. The objective function in (24) is linear in $\mathbf{W}$ for a given $\mathbf{z}$ and linear in $\mathbf{z}$ when $\mathbf{W}$ is fixed. Additionally, the constraint sets for both $\mathbf{W}$ and $\mathbf{z}$ are compact and convex. Therefore, by applying the minimax theorem Sion (1958), we can interchange the max and min operators in (24), yielding the following equivalent problem:

$$
\begin{aligned}
\min_{\{z_{ij}\}} \max_{\mathbf{W}} \quad & 2\operatorname{Tr}\left(\mathbf{A}^T \mathbf{W}\right) + \sum_{1 \leq i < j \leq C} z_{ij} c_{ij} \\
\text{s.t.} \quad & z_{ij} \geq 0, \quad \sum_{1 \leq i < j \leq C} z_{ij} = 1 \\
& \begin{bmatrix} \mathbf{I}_{d'} & \mathbf{W}^T \\ \mathbf{W} & \mathbf{I}_d \end{bmatrix} \succcurlyeq 0
\end{aligned}
\quad .
\tag{25}
$$

The inner maximization problem in (25) can be directly solved in closed form. Focusing on the first term in the objective of (24), we can apply the Von Neumann inequality Marshall (1979), yielding:

$$
\operatorname{Tr}\left(\mathbf{A}^T \mathbf{W}\right) \leq \sum_{k=1}^{d'} \sigma_k(\mathbf{A}) \sigma_k(\mathbf{W})
\tag{26}
$$

where $\sigma_k(\mathbf{A})$ and $\sigma_k(\mathbf{W})$ represent the non-zero singular values of $\mathbf{A}$ and $\mathbf{W}$, respectively. Since $\sigma_k(\mathbf{W}) \leq 1$, it follows that

$$
\operatorname{Tr}\left(\mathbf{A}^T \mathbf{W}\right) \leq \sum_{k=1}^{d'} \sigma_k(\mathbf{A}),
\tag{27}
$$

with the equality obtained for

$$
\mathbf{W}^* = \mathbf{A}\left(\mathbf{A}^T \mathbf{A}\right)^{-\frac{1}{2}}.
\tag{28}
$$

Indeed,

$$
\operatorname{Tr}\left(\mathbf{A}^T \mathbf{W}^*\right) = \operatorname{Tr}\left(\left(\mathbf{A}^T \mathbf{A}\right)\left(\mathbf{A}^T \mathbf{A}\right)^{-\frac{1}{2}}\right) = \operatorname{Tr}\left(\left(\mathbf{A}^T \mathbf{A}\right)^{\frac{1}{2}}\right) = \sum_{i=1}^{d'} \sigma_i(\mathbf{A}).
\tag{29}
$$

---
**Algorithm 1** MM4MM for WCCS (SDP approach)

---
**Input:** Initial estimate $\mathbf{W}^0, \{\tilde{\mathbf{S}}_{Cij}\}$ for $1 \leq i < j \leq C$, and convergence threshold $\epsilon = 10^{-5}$.
Set $t = 0$.

1: **repeat**
2:     Compute $\{\mathbf{A}_{ij}, c_{ij}\}$ from (21), (22).
3:     Compute $\mathbf{z}^*$ by solving (30).
4:     Obtain $\mathbf{W}^{t+1}$ from (31).
5:     Set $t = t + 1$.
6: **until** $\frac{\|\mathbf{W}^{t+1} - \mathbf{W}^t\|}{\|\mathbf{W}^t\|} \leq \epsilon$
7: **Output:** $\mathbf{W}^* = \mathbf{W}^t$ at convergence.

---

Notice that $\mathbf{W}^*$ fulfills the constraint in (16), i.e., $(\mathbf{W}^*)^T \mathbf{W}^* = \mathbf{I}$. Therefore, as asserted, the maximizer of (23) satisfies the constraint $\mathbf{W}^T \mathbf{W} = \mathbf{I}$ at each iteration. Substituting (28) into (25) leads to the following problem:

$$
\begin{aligned}
\min_{\{z_{ij}\}} \quad & 2 \sum_{i=1}^{d'} \sigma_i(\mathbf{A}(\mathbf{z})) + \sum_{1 \leq i < j \leq C} z_{ij} c_{ij} \\
\text{s.t.} \quad & z_{ij} \geq 0, \sum_{1 \leq i < j \leq C} z_{ij} = 1
\end{aligned}
\tag{30}
$$

where we emphasize that $\mathbf{A}$ depends on $\mathbf{z}$. The first term in (30) equals twice the nuclear norm of $\mathbf{A}(\mathbf{z})$, represented as $\|\mathbf{A}(\mathbf{z})\|_*$, which is a convex function of $\mathbf{A}$ and thus $\{z_k\}$. As a result, (30) is a convex problem, similar to (23), and can be reformulated as an SDP Recht et al. (2010). However, compared to (23), the number of variables and constraints in (30) is smaller, offering a potential advantage.

After obtaining the minimizer $\mathbf{z}^*$ by solving (30), the corresponding $\mathbf{W}$ (which is the maximizer of (23)) can be calculated as:

$$
\mathbf{W}^{(t+1)} = \mathbf{A}(\mathbf{z}^*) \left(\mathbf{A}^T(\mathbf{z}^*) \mathbf{A}(\mathbf{z}^*)\right)^{-\frac{1}{2}},
\tag{31}
$$

and it will be used as the next iteration. The MM procedure iterates this process until convergence is achieved. The steps of the MM4MM for the WCCS problem are summarized in Algorithm 1.

The primary computational demand of the proposed algorithm stems from calculating $\{\mathbf{A}_{ij}\}$ for $1 \leq i < j \leq C$, solving the SDP in (30), and updating $\mathbf{W}^{t+1}$ as outlined in (31). The computation of $\{\mathbf{A}_{ij}\}$ for $1 \leq i < j \leq C$ involves matrix-matrix multiplications, which can be performed in $\mathcal{O}(\frac{C(C-1)d'd^2}{2})$ operations. Solving the SDP in (30) has a computational cost of approximately $\mathcal{O}((d + d')^{4.5})$ operations, while the evaluation in (31) requires $\mathcal{O}(dd'^2) + \mathcal{O}(d'^3)$ operations. Therefore, the total computational complexity per iteration is $\mathcal{O}((d + d')^{4.5})$.

In the following, we introduce an alternative method for solving (30) aimed at reducing the computational load. To achieve this, we begin with the formulation presented in (30):

$$
\begin{aligned}
\min_{\{z_{ij}\}} \quad & 2\|\mathbf{A}(\mathbf{z})\|_* + \sum_{1 \leq i < j \leq C} z_{ij} c_{ij} \\
\text{s.t.} \quad & z_{ij} \geq 0, \sum_{1 \leq i < j \leq C} z_{ij} = 1
\end{aligned}
\tag{32}
$$

Let us introduce an auxillary variable $\mathbf{\Phi}(\mathbf{\Phi} \succ \mathbf{0})$ and reformulate (32) as follows:

$$
\begin{aligned}
\min_{\{z_{ij}\}, \mathbf{\Phi} \succ 0} \quad & \text{Tr}\left(\mathbf{\Phi}^{-1}\right) + \text{Tr}\left(\mathbf{A}^T(\mathbf{z})\mathbf{A}(\mathbf{z})\mathbf{\Phi}^t\right) + \sum_{1 \leq i < j \leq C} z_{ij} c_{ij} \\
\text{s.t.} \quad & z_{ij} \geq 0, \sum_{1 \leq i < j \leq C} z_{ij} = 1
\end{aligned}
\tag{33}
$$

The problems (32) and (33) are equivalent, as shown below. By minimizing (33) with respect to $\mathbf{\Phi}$ while keeping $\mathbf{z}$ fixed, we find that the minimizer is $\mathbf{\Phi}^* = \left(\mathbf{A}^T \mathbf{A}\right)^{-\frac{1}{2}}$. Substituting this back into (33) results in

---

**Algorithm 2** Alternating Minimization Approach for Solving (30)

---

1: **Input:** Initial estimate $\mathbf{z}^0$, coefficients $\{c_{ij}, \mathbf{A}_{ij}\}$, and convergence threshold $\epsilon = 10^{-5}$
2: Set $t = 0$
3: **repeat**
4:   Compute $\mathbf{\Phi}^t = \left(\mathbf{A}^T(\mathbf{z}^t)\mathbf{A}(\mathbf{z}^t)\right)^{-\frac{1}{2}}$
5:   Obtain $\mathbf{z}^{t+1}$ by solving equation (35)
6:   Set $t = t + 1$
7: **until** $\frac{\|\mathbf{z}^{t+1} - \mathbf{z}^t\|}{\|\mathbf{z}^t\|} \leq \epsilon$
8: **Output:** $\mathbf{z}^* = \mathbf{z}^t$ at convergence

---

the problem (32). The problem in (33) can be tackled using an alternating minimization method: for a given $\mathbf{z}$, denoted as $\mathbf{z}^t$, we obtain the minimizer $\mathbf{\Phi}^t$ as described above. With $\mathbf{\Phi} = \mathbf{\Phi}^t$ fixed, the minimizer $\mathbf{z}$ can be found by solving the following convex problem:

$$
\begin{aligned}
\min_{\{z_{ij}\}} \quad & \mathrm{Tr}\left(\mathbf{A}^T(\mathbf{z})\mathbf{A}(\mathbf{z})\mathbf{\Phi}^t\right) + \sum_{1 \leq i < j \leq C} z_{ij} c_{ij} \\
\text{s.t.} \quad & z_{ij} \geq 0, \sum_{1 \leq i < j \leq C} z_{ij} = 1
\end{aligned}
\tag{34}
$$

which can be reformulated as a quadratic program (QP):

$$
\begin{aligned}
\min_{\{z_{ij}\}} \quad & \mathbf{z}^T \mathbf{Q} \mathbf{z} + \sum_{1 \leq i < j \leq C} z_{ij} c_{ij} \\
\text{s.t.} \quad & z_{ij} \geq 0, \sum_{1 \leq i < j \leq C} z_{ij} = 1
\end{aligned}
\tag{35}
$$

where

$$
\mathbf{Q} = \mathbf{S}^{\mathrm{T}}(\mathbf{I} \otimes \mathbf{\Phi}^t)\mathbf{S},
\tag{36}
$$

with $\mathbf{S} = [\boldsymbol{v}_1 \ \boldsymbol{v}_2 \ \cdots \ \boldsymbol{v}_K]$, $K = \frac{C(C-1)}{2}$, and $\boldsymbol{v}_k = \mathrm{vec}(\mathbf{A}_{ij})$, where $k = \frac{(i-1)(2C-i)}{2} + j - i$. Here, $\otimes$ denotes the Kronecker product. For the derivation of the relationship between $\mathbf{Q}$ and the matrices $\mathbf{A}_{ij}$, refer to Appendix 6.2. The quadratic program (QP) in (35) can be efficiently solved using standard solvers, with a computational complexity that includes updating $\mathbf{\Phi}^t$ at a cost of approximately $\mathcal{O}\left(C^6\right) + \mathcal{O}\left(d^3\right)$ operations. This is significantly lower than $\mathcal{O}\left((d+d')^{4.5}\right)$. However, unlike the problem in (30), the QP in (35) must be solved multiple times for various $\mathbf{\Phi}^t$ matrices. Fortunately, this typically requires only a few iterations (usually fewer than 10). The iterative steps for the alternating minimization algorithm used to find the solution $\mathbf{z}^*$ for (30) are outlined in Algorithm 2.

### 3.4 Solving the WCCS problem under the sparsity penalty

In this subsection, we present an extension of the proposed method to solve problem (4):

$$
\begin{aligned}
\max_{\mathbf{W}} \min_{1 \leq i < j \leq C} \quad & \{f_{ij}(\mathbf{W}) - \lambda \|\mathbf{W}\|_1\} \\
\text{s.t.} \quad & \mathbf{W}^T \mathbf{W} = \mathbf{I}
\end{aligned}
\tag{37}
$$

where $\lambda$ is a specified parameter that governs the sparsity of $\mathbf{W}$. In a manner similar to (17), we relax the semi-orthogonality constraint:

$$
\begin{aligned}
\max_{\mathbf{W}} \min_{1 \leq i < j \leq C} \quad & f_{ij}(\mathbf{W}) - \lambda \|\mathbf{W}\|_1 \\
\text{s.t.} \quad & \begin{bmatrix} \mathbf{I}_{d'} & \mathbf{W}^T \\ \mathbf{W} & \mathbf{I}_d \end{bmatrix} \succcurlyeq 0
\end{aligned}
\tag{38}
$$

For a fixed $\mathbf{W} = \mathbf{W}^t$, the quadratic functions $\{f_{ij}(\mathbf{W})\}$ can be minorized using their tangent hyperplanes, leading to the following surrogate problem:

$$
\max_{\mathbf{W}} \min_{1 \leq i < j \leq C} \quad \{g_{ij}(\mathbf{W}) - \lambda \|\mathbf{W}\|_1\}
$$
$$
\text{s.t.} \qquad \begin{bmatrix} \mathbf{I}_{d'} & \mathbf{W}^T \\ \mathbf{W} & \mathbf{I}_d \end{bmatrix} \succcurlyeq 0 \tag{39}
$$

where, as previously mentioned, $g_{ij}(\mathbf{W}) = 2\,\mathrm{Tr}\left(\mathbf{A}_{ij}^T \mathbf{W}\right) + c_{ij}$. It is obvious that (39) is convex (specifically an SDP) and can be solved using CVX. Similar to subsection 3.3, the optimal solution of (39) can be shown to satisfy the constraint $\mathbf{W}^T\mathbf{W} = \mathbf{I}$. To demonstrate this, we reformulate the problem in (39) using auxiliary variables $\mathbf{B}$ and $\mathbf{z}$ as follows:

$$
\max_{\mathbf{W}} \min_{\{z_{ij}\}, \mathbf{B}} \quad \sum_{1 \leq i < j \leq C} z_{ij} g_{ij}(\mathbf{W}) + \lambda\,\mathrm{Tr}\left(\mathbf{B}^T \mathbf{W}\right)
$$
$$
\text{s.t.} \qquad z_{ij} \geq 0, \sum_{1 \leq i < j \leq C} z_{ij} = 1
$$
$$
\begin{bmatrix} \mathbf{I}_{d'} & \mathbf{W}^T \\ \mathbf{W} & \mathbf{I}_d \end{bmatrix} \succcurlyeq \mathbf{0} \tag{40}
$$
$$
|[\mathbf{B}]_{ij}| \leq 1 \quad \forall i, j
$$

Minimizing (40) with respect to $\mathbf{B}$ and $\mathbf{z}$ yields the objective in (39), thus (39) and (40) are equivalent. By applying the minimax theorem (as the objective and constraints of (40) meet the required conditions), the max and min operators can be interchanged:

$$
\min_{\{z_{ij}\}, \mathbf{B}} \max_{\mathbf{W}} \quad \sum_{1 \leq i < j \leq C} z_{ij} g_{ij}(\mathbf{W}) + \lambda\,\mathrm{Tr}\left(\mathbf{B}^T \mathbf{W}\right)
$$
$$
\text{s.t.} \qquad z_{ij} \geq 0, \sum_{1 \leq i < j \leq C} z_{ij} = 1
$$
$$
\begin{bmatrix} \mathbf{I}_{d'} & \mathbf{W}^T \\ \mathbf{W} & \mathbf{I}_d \end{bmatrix} \succcurlyeq \mathbf{0} \tag{41}
$$
$$
|[\mathbf{B}]_{ij}| \leq 1 \quad \forall i, j
$$

By substituting the expression $g_{ij}(\mathbf{W}) = 2\,\mathrm{Tr}\left(\mathbf{A}_{ij}^T \mathbf{W}\right) + c_{ij}$, (41) can be reformulated as:

$$
\min_{\{z_{ij}\}, \mathbf{B}} \max_{\mathbf{W}} \quad 2\,\mathrm{Tr}\left(\left(\mathbf{A} + \frac{\lambda}{2}\mathbf{B}\right)^T \mathbf{W}\right) + \sum_{1 \leq i < j \leq C} z_{ij} c_{ij}
$$
$$
\text{s.t.} \qquad z_{ij} \geq 0, \sum_{1 \leq i < j \leq C} z_{ij} = 1
$$
$$
\begin{bmatrix} \mathbf{I}_{d'} & \mathbf{W}^T \\ \mathbf{W} & \mathbf{I}_d \end{bmatrix} \succcurlyeq 0 \tag{42}
$$
$$
|[\mathbf{B}]_{ij}| \leq 1 \quad \forall i, j
$$

Similar to (28), the maximizer $\mathbf{W}$ of (42) can be obtained in closed form:

$$
\mathbf{W}^* = \left(\mathbf{A} + \frac{\lambda}{2}\mathbf{B}\right)\left(\left(\mathbf{A} + \frac{\lambda}{2}\mathbf{B}\right)^T \left(\mathbf{A} + \frac{\lambda}{2}\mathbf{B}\right)\right)^{-\frac{1}{2}}, \tag{43}
$$

which meets the constraint $\mathbf{W}^T\mathbf{W} = \mathbf{I}$. Therefore, in this case as well, the MM iterations meet the semi-orthogonality constraint. The pseudocode for the proposed approach with sparsity penalty is summarized in Algorithm 3.

---

**Algorithm 3** MM4MM for WCCS with sparsity penalty

---

**Input:** Initial estimate $\mathbf{W}^0$, set of matrices $\left\{\tilde{\mathbf{S}}_{Cij}\right\}$ for $1 \leq i < j \leq C$, penalty parameter $\lambda$, and convergence threshold $\epsilon = 10^{-5}$.
**Initialize:** Set $t = 0$.

1: **repeat**
2:      Compute $\{\mathbf{A}_{ij}, c_{ij}\}$ from (21), (22).
3:      Obtain $\mathbf{W}^{t+1}$ by solving (39).
4:      Set $t = t + 1$.
5: **until** $\frac{\|\mathbf{W}^{t+1} - \mathbf{W}^t\|}{\|\mathbf{W}^t\|} \leq \epsilon$
**Output:** $\mathbf{W}^* = \mathbf{W}^t$ at convergence.

---

## 4 Numerical results

### 4.1 Datasets

We evaluate the performance of the MM4MM algorithm for solving WCCS problem, with and without sparsity penalty. The evaluation is conducted on six real-world datasets from the UCI Machine Learning Repository and Kaggle. These datasets are briefly described below:

The Iris dataset consists of 150 instances, each represented by four features: sepal length, sepal width, petal length, and petal width. The label categorizes each instance into one of three classes: Iris-setosa, Iris-versicolor, or Iris-virginica, with 50 samples per class. The Wine dataset contains 177 instances, each described by 13 chemical attributes such as alcohol content, malic acid, and ash, with labels indicating the type of wine corresponding to one of three cultivars. The Seeds dataset comprises 210 instances with seven features that describe the geometric properties of wheat kernels, including area, perimeter, and compactness, and is divided into three classes representing different types of wheat: Kama, Rosa, and Canadian. The Prestige dataset includes 98 instances with features representing education level, income, percentage of women in the occupation, and prestige scores. Depending on the analysis, the labels can be used for classification or regression tasks. The Diamonds dataset contains 599 instances and four main features: carat weight, depth, table size, and clarity. The labels represent the quality of the cut, categorized into four classes: Fair, Good, Ideal, and Premium. Finally, the Digit dataset Prabhu (2019) is a high-dimensional dataset of 1,797 samples, each represented by 64-dimensional features corresponding to images of handwritten digits (0 through 9).

### 4.2 Methods

For the five non-Digit datasets (Iris, Wine, Seeds, Prestige, Diamond), we performed five-fold cross-validation: each dataset was divided into five subsets, with four subsets used for training and one for testing per fold. Performance metrics were computed over the five test folds. For the high-dimensional Digit dataset, we followed Wang et al. (2024), performing 20 independent experimental runs where 50% of samples were randomly selected for training and the remainder for testing in each run. PCA preprocessing was applied to all datasets following Wang et al. (2024); Su et al. (2015), preserving 98% of the variance. Final results report average accuracy ± standard deviation (over five folds for non-Digit datasets; over 20 runs for Digit). For comparison, we included several widely used discriminant analysis methods: LDA Fisher (1936); Rao (1948), HLDA Loog & Duin (2004), MMDA Bian & Tao (2011), WHMMDA Su et al. (2018; 2015), and MMRA Wang et al. (2024).

For all datasets, the original dimensionality $d$ was reduced to various potential values from 1 to $d-1$, except for LDA, where the maximum dimensionality of the selected subspace was constrained to $C-1$ to achieve its best performance and allow for a fair comparison across methods.

Classification in the reduced subspaces was performed using three classifiers: the nearest neighbor classifier (1-NN), the nearest mean classifier (NM), and the quadratic discriminant analysis (QDA). The quadratic

Table 1: Classifier Results Across 6 Datasets

| Dataset | Classifier | LDA | HLDA | MMDA | WHMMDA | MMRA | MM4MM (QP) | MM4MM (Sparse) |
|---|---|---|---|---|---|---|---|---|
| Iris | 1-NN | 0.0667 (2, Std: 0.0236) | **0.0533** (3, Std: 0.0298) | 0.0667 (2, Std: 0.0471) | 0.0867 (3, Std: 0.0298) | **0.0533** (3, Std: 0.0298) | 0.0600 (3, Std: 0.0494) | **0.0533** (3, Std: 0.0380) |
| | NM | 0.0533 (2, Std: 0.0298) | 0.0467 (3, Std: 0.0447) | 0.0267 (3, Std: 0.0279) | 0.0267 (3, Std: 0.0149) | 0.0333 (3, Std: 0.0333) | **0.0200** (3, Std: 0.0183) | **0.0200** (3, Std: 0.0183) |
| | QDA | 0.0467 (2, Std: 0.0380) | 0.0400 (3, Std: 0.0279) | 0.0333 (3, Std: 0.0236) | 0.0333 (3, Std: 0.0236) | 0.0400 (3, Std: 0.0279) | 0.0333 (3, Std: 0.0236) | **0.0267** (3, Std: 0.0279) |
| Wine | 1-NN | 0.0224 (2, Std: 0.0125) | 0.0170 (12, Std: 0.0155) | 0.0168 (3, Std: 0.0154) | 0.0279 (4, Std: 0.0481) | **0.0056** (2, Std: 0.0124) | 0.0225 (10, Std: 0.0126) | 0.0170 (3, Std: 0.0255) |
| | NM | 0.0113 (2, Std: 0.0154) | 0.0113 (11, Std: 0.0154) | 0.0224 (4, Std: 0.0125) | 0.0171 (8, Std: 0.0256) | 0.0225 (12, Std: 0.0238) | 0.0168 (10, Std: 0.0154) | **0.0057** (12, Std: 0.0128) |
| | QDA | 0.0113 (2, Std: 0.0154) | 0.0168 (12, Std: 0.0154) | 0.0113 (11, Std: 0.0154) | 0.0222 (4, Std: 0.0232) | 0.0057 (8, Std: 0.0128) | **0.0056** (12, Std: 0.0124) | **0.0056** (12, Std: 0.0124) |
| Seeds | 1-NN | 0.0571 (2, Std: 0.0213) | 0.0476 (6, Std: 0.0238) | **0.0381** (3, Std: 0.0319) | 0.0667 (5, Std: 0.0391) | 0.3667 (6, Std: 0.0764) | 0.0524 (4, Std: 0.0391) | 0.0571 (4, Std: 0.0213) |
| | NM | 0.0429 (2, Std: 0.0199) | **0.0333** (6, Std: 0.0130) | 0.0381 (6, Std: 0.0271) | 0.0381 (6, Std: 0.0130) | 0.3190 (6, Std: 0.1124) | **0.0333** (6, Std: 0.0271) | **0.0333** (4, Std: 0.0319) |
| | QDA | 0.0381 (2, Std: 0.0271) | 0.0381 (4, Std: 0.0130) | 0.0381 (5, Std: 0.0213) | 0.0333 (4, Std: 0.0213) | 0.3429 (6, Std: 0.1099) | 0.0333 (3, Std: 0.0213) | **0.0286** (4, Std: 0.0391) |
| Prestige | 1-NN | 0.0716 (2, Std: 0.0284) | 0.0811 (1, Std: 0.0575) | 0.0911 (4, Std: 0.0406) | 0.1021 (4, Std: 0.0708) | 0.3442 (4, Std: 0.1944) | 0.0632 (3, Std: 0.0942) | **0.0400** (4, Std: 0.0548) |
| | NM | 0.0826 (2, Std: 0.0484) | 0.0916 (4, Std: 0.0650) | 0.0805 (4, Std: 0.0567) | 0.0811 (4, Std: 0.0665) | 0.3863 (4, Std: 0.1916) | 0.0842 (4, Std: 0.1212) | **0.0716** (3, Std: 0.0780) |
| | QDA | 0.0721 (2, Std: 0.0303) | 0.0816 (4, Std: 0.0572) | 0.0705 (4, Std: 0.0444) | 0.0811 (4, Std: 0.0762) | 0.3458 (4, Std: 0.1833) | 0.0721 (4, Std: 0.0596) | **0.0526** (3, Std: 0.0912) |
| Diamond | 1-NN | 0.0484 (2, Std: 0.0162) | 0.0518 (3, Std: 0.0138) | 0.0484 (2, Std: 0.0170) | 0.1387 (3, Std: 0.0463) | 0.0485 (3, Std: 0.0293) | 0.1220 (3, Std: 0.0607) | **0.0467** (3, Std: 0.0172) |
| | NM | 0.0835 (3, Std: 0.0405) | 0.0768 (3, Std: 0.0148) | 0.0852 (3, Std: 0.0194) | 0.1654 (3, Std: 0.0469) | 0.0769 (3, Std: 0.0453) | 0.1403 (3, Std: 0.0611) | **0.0752** (3, Std: 0.0266) |
| | QDA | 0.0318 (3, Std: 0.0217) | **0.0217** (3, Std: 0.0127) | 0.0351 (3, Std: 0.0161) | 0.1236 (3, Std: 0.0509) | 0.0300 (3, Std: 0.0173) | 0.0919 (3, Std: 0.0562) | **0.0217** (3, Std: 0.0112) |
| Digit | 1-NN | 0.0777 (6, Std: 0.0081) | **0.0181** (36, Std: 0.0045) | 0.0454 (11, Std: 0.0063) | 0.0335 (16, Std: 0.0055) | 0.0222 (46, Std: 0.0222) | 0.0299 (16, Std: 0.0043) | 0.0299 (16, Std: 0.0043) |
| | NM | 0.0730 (6, Std: 0.0070) | 0.0483 (41, Std: 0.0072) | 0.0484 (26, Std: 0.0049) | 0.0495 (46, Std: 0.0058) | 0.0571 (46, Std: 0.0072) | **0.0476** (41, Std: 0.0073) | **0.0476** (41, Std: 0.0073) |
| | QDA | 0.0658 (6, Std: 0.0070) | 0.0231 (21, Std: 0.0050) | 0.0275 (36, Std: 0.0069) | 0.0252 (31, Std: 0.0047) | 0.0553 (46, Std: 0.0093) | **0.0223** (26, Std: 0.0060) | **0.0223** (26, Std: 0.0060) |

classifier utilized the following decision rule:

$$\hat{i} = \arg \min_{i=1,\ldots,C} \left\{ (\mathbf{x} - \mathbf{m}_i)^T \boldsymbol{\Sigma}_i^{-1} (\mathbf{x} - \mathbf{m}_i) + \log |\boldsymbol{\Sigma}_i| \right\},$$
$$\mathbf{x} \in C_{\hat{i}}$$

where $\mathbf{m}_i$ represents the mean vector of class $i$, and $\boldsymbol{\Sigma}_i$ is the covariance matrix of class $i$. This ensured that the classifiers could capture both linear and non-linear separability, providing a thorough evaluation of performance across the different dimensionality reduction methods. To refer to our approaches in the experimental results, we name them MM4MM (QP) and MM4MM (Sparse). It is important to highlight that for MM4MM (Sparse), we performed a grid search over $\lambda \in \{0.001, , 0.1, , 0.2, \ldots, 1.0\}$ and, for each value, computed the objective function according to the optimal solution formulation. We then selected the $\lambda$ that yielded the highest objective.

### 4.3 Results

Tables 1 presents the optimal average classification error rates along with their corresponding standard deviations and dimensionalities for the various dimensionality reduction methods across the five datasets. It is worth mentioning that because each method—including ours and the baselines—reaches its peak accuracy at a distinct target dimensionality, we follow the evaluation protocol from Su et al. (2015) and report results at each method's respective optimal dimensionality. As a result, while Section 3 previously outlined the computational complexity of our approach, making a direct runtime comparison would be unfair.

As we can see from Table 1, for the Iris dataset, the proposed approaches consistently outperform traditional methods for non-linear classifiers. The NM classifier, paired with either MM4MM (QP) or MM4MM (Sparse), achieves the lowest average error rates. Furthermore, in both the 1-NN and QDA classifiers, the best approach is found to be MM4MM (Sparse), demonstrating the strength of combining our advanced dimensionality reduction method with all the classifiers mentioned.

On the Wine dataset, which features complex chemical attributes, MM4MM (Sparse) demonstrates significant improvements in minimizing error rates, particularly with the NM and QDA classifiers, as well as MM4MM (QP) with the QDA classifier.

The results from the Seeds dataset reveal that, although the MMDA approach outperforms others for the 1-NN classifier, the MM4MM (Sparse) method achieves the lowest error rates for the NM and QDA classifiers, highlighting its superior classification performance. Additionally, for the NM classifier, both MM4MM (QP) and HLDA demonstrate competitive performance, comparable to the MM4MM methods. Furthermore, for the QDA classifier, MM4MM (QP) and WHMMDA show the second-best performance.

For the Prestige dataset, which consists of socio-economic data, the proposed methods—MM4MM (Sparse) as the best and MM4MM (QP) as the second best—outperform traditional approaches in terms of classification error and standard deviations across all three classifiers. This result highlights the robustness of MM4MM (Sparse) in optimizing subspace transformations.

The Diamonds dataset, used for modeling regression scenarios, demonstrates that the QDA classifier combined with MM4MM (Sparse) achieves the lowest error rates, confirming the efficacy of this approach for handling datasets with complex feature inter-dependencies. In contrast, all competing methods (except HLDA with the QDA classifier) show higher error rates, highlighting their diminished effectiveness in non-linear data environments.

For the high-dimensional Digit dataset, our proposed MM4MM (QP) method demonstrates superior effectiveness in 2 out of 3 classifiers. While HLDA achieves the lowest error rate under the 1-NN classifier, MM4MM (QP) outperforms all competing methods under both the NM and QDA classifiers. This consistent top-tier performance across multiple classifiers confirms MM4MM (QP) as a robust solution for high-dimensional data, yielding the lowest error rates in the majority of experimental settings. It is worth mentioning that the reason why MM4MM (Sparse) shows similar values to MM4MM (QP) is that its best performance is achieved at lambda = 0.001 for this dataset.

## 5    Conclusion

In this work, we introduced a new discriminative feature learning method built on a minorization-maximization framework for min-max (MM4MM), aimed specifically at addressing the problem of "worst-case class separation (WCCS)". The algorithm was designed to operate using a relaxed semi-orthogonality constraint, which was shown to be tight at every iteration.

Our approach began with a vanilla version that required solving a semi-definite program (SDP) at each iteration. To simplify this, we also developed a method that reduced the problem to a quadratic program by constructing the dual of the surrogate maximization problem. Additionally, we proposed a reformulation of the WCCS problem that includes a sparsity penalty.

The proposed algorithms are computationally efficient and enjoy guaranteed convergence. A key advantage of the proposed approach is that it does not require any hyperparameter tuning, except for the sparsity-based version where users need to select a penalty parameter for controlling sparsity. Experiments conducted on multiple machine learning datasets demonstrated the strong performance of the MM4MM approach.

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

# 6   Appendix

## 6.1   The proof of Lemma 1

*Proof.* Note that in (16) (for any $i, j$)

$$\text{Tr}\left(\mathbf{W}^T \tilde{\mathbf{S}}_{Cij} \mathbf{W}\right) = \text{Tr}\left(\tilde{\mathbf{S}}_{Cij} \mathbf{W}\mathbf{W}^T\right) = \text{Tr}\left(\tilde{\mathbf{S}}_{Cij} \mathbf{W}\mathbf{Q}\mathbf{Q}^T\mathbf{W}^T\right),$$

which holds for any matrix $\mathbf{Q}$ such that $\mathbf{Q}\mathbf{Q}^T = \mathbf{I}$. Therefore, for any matrix $\mathbf{W}$ that meets the relaxed constraint, we can select $\mathbf{Q}$ such that $\mathbf{Q}^T\mathbf{W}^T\mathbf{W}\mathbf{Q} = \text{diagonal} \triangleq \mathbf{\Lambda} \preccurlyeq \mathbf{I}$. Since $\mathbf{W}^T\mathbf{W}$ and $\mathbf{W}\mathbf{W}^T$ share the same non-zero eigenvalues, it follows that:

$$\mathbf{W}\mathbf{W}^T = \mathbf{V}\mathbf{\Lambda}\mathbf{V}^T, \tag{44}$$

where $\mathbf{V}$ contains the principal eigenvectors of $\mathbf{W}\mathbf{W}^T$ and $\mathbf{V}^T\mathbf{V} = \mathbf{I}$. Using (44) we have:

$$\begin{aligned}
\text{Tr}\left(\mathbf{W}^T\tilde{\mathbf{S}}_{Cij}\mathbf{W}\right) &= \text{Tr}\left(\left(\mathbf{V}^T\tilde{\mathbf{S}}_{Cij}\mathbf{V}\right)\mathbf{\Lambda}\right) \\
&= \sum_{k=1}^{d'}\left(\mathbf{V}^T\tilde{\mathbf{S}}_{Cij}\mathbf{V}\right)_{kk}\mathbf{\Lambda}_{kk} \leq \sum_{k=1}^{d'}\left(\mathbf{V}^T\tilde{\mathbf{S}}_{Cij}\mathbf{V}\right)_{kk}.
\end{aligned} \tag{45}$$

Hence, all functions in (16) take larger values when $\mathbf{\Lambda} = \mathbf{I}$ than when $\mathbf{\Lambda} \preccurlyeq \mathbf{I}$. This indicates that the global maximizer of (16), constrained by $\mathbf{W}^T\mathbf{W} \preccurlyeq \mathbf{I}$, will indeed satisfy the constraint in (16). Therefore, the relaxation $\mathbf{W}^T\mathbf{W} \preccurlyeq \mathbf{I}$ does not alter the solution of (16), and the proof of Lemma 1 is completed. □

## 6.2   Proof of (36)

*Proof.* To find an explicit form of $\mathbf{Q}$, let us define $\tilde{\mathbf{A}}_k$ and $\tilde{z}_k$ as $\tilde{\mathbf{A}}_k \triangleq \mathbf{A}_{ij}$, $\tilde{z}_k \triangleq z_{ij}$ where $k = \frac{(i-1)(2C-i)}{2} + j - i$. As a result $\mathbf{A}$ can be rewritten as $\mathbf{A} = \sum_{k=1}^{K} \tilde{z}_k \tilde{\mathbf{A}}_k$ where $K = \frac{C(C-1)}{2}$. Calculating $\mathbf{A}^T\mathbf{A}$ gives:

$$\mathbf{A}^T\mathbf{A} = \left(\sum_{k=1}^{K}\tilde{z}_k\tilde{\mathbf{A}}_k\right)^T\left(\sum_{l=1}^{K}\tilde{z}_l\tilde{\mathbf{A}}_l\right) = \sum_{k=1}^{K}\sum_{l=1}^{K}\tilde{z}_k\tilde{z}_l\tilde{\mathbf{A}}_k^T\tilde{\mathbf{A}}_l.$$

Multiplying both sides of the above equation by $\mathbf{\Phi}$ from the right and taking the trace, we obtain:

$$\text{Tr}\left(\mathbf{A}^T\mathbf{A}\mathbf{\Phi}^t\right) = \sum_{k=1}^{K}\sum_{l=1}^{K}\tilde{z}_k\tilde{z}_l\,\text{Tr}\left(\tilde{\mathbf{A}}_k^T\tilde{\mathbf{A}}_l\mathbf{\Phi}^t\right). \tag{46}$$

On the other hand, the quadratic form $\mathbf{z}^T\mathbf{Q}\mathbf{z}$ expands to:

$$\mathbf{z}^T\mathbf{Q}\mathbf{z} = \sum_{k=1}^{K}\sum_{l=1}^{K}\tilde{z}_k Q_{k,l}\tilde{z}_l. \tag{47}$$

By comparing (46) and (47), we obtain:

$$Q_{k,l} = \text{Tr}\left(\tilde{\mathbf{A}}_k^T\tilde{\mathbf{A}}_l\mathbf{\Phi}^t\right).$$

Using the vectorization operator, we define $\boldsymbol{v}_k = \text{vec}(\tilde{\mathbf{A}}_k)$. The trace property gives:

$$\text{Tr}\left(\tilde{\mathbf{A}}_k^T\tilde{\mathbf{A}}_l\mathbf{\Phi}^t\right) = \boldsymbol{v}_k^T\left(\mathbf{I} \otimes \mathbf{\Phi}^t\right)\boldsymbol{v}_l,$$

where $\mathbf{I}$ is the identity matrix and $\otimes$ denotes the Kronecker product. We construct $\mathbf{S}$ by stacking $\boldsymbol{v}_k$ as columns:

$$\mathbf{S} = [\boldsymbol{v}_1\ \boldsymbol{v}_2\ \cdots\ \boldsymbol{v}_K].$$

Thus, we have:

$$\mathrm{Tr}\left(\mathbf{A}^{\mathrm{T}}\mathbf{A}\mathbf{\Phi}^t\right) = \mathbf{z}^{\mathrm{T}}\mathbf{S}^{\mathrm{T}}\left(\mathbf{I}\otimes\mathbf{\Phi}^t\right)\mathbf{S}\mathbf{z}. \tag{48}$$

Equating (48) to $\mathbf{z}^{\mathrm{T}}\mathbf{Q}\mathbf{z}$, we derive:

$$\mathbf{Q} = \mathbf{S}^{\mathrm{T}}\left(\mathbf{I}\otimes\mathbf{\Phi}^t\right)\mathbf{S},$$

where $\mathbf{S} = [\boldsymbol{v}_1\ \boldsymbol{v}_2\ \ldots\ \boldsymbol{v}_K]$ and $\boldsymbol{v}_k = \mathrm{vec}(\mathbf{A}_{ij})$ with:

$$k = \frac{(i-1)(2C-i)}{2} + j - i,$$

and the proof is completed. $\qquad\square$

