# OpenReview forum: "A Max-Min Approach to the Worst-Case Class Separation Problem"
_TMLR — Accepted by TMLR_

### Review · Reviewer_cPL4 · 2025-03-19

**Summary Of Contributions:**

This paper addresses the worst-case class separation problem (WCSP) within the context of feature discriminant learning. The WCSP involves maximizing the minimum of a convex objective while adhering to a nonconvex semi-orthogonality constraint. The authors propose an algorithmic framework based on two main ideas. First, they suggest relaxing the semi-orthogonality constraint into a convex one, ensuring that the global maximizer of the relaxed problem still meets the original non-convex constraint. Second, they tackle the optimization of the objective function—which features a max-min operation—using a minorization-maximization approach.

Although the resulting semi-definite optimization problem is convex, its computational complexity remains high. To address this, the authors provide an alternative quadratic formulation that can be efficiently solved with standard algorithms. Additionally, they explore a sparse version of the WCSP and present a relaxation that is also suitable for convex solving. Comparative experiments conducted on several datasets demonstrate the effectiveness of this framework.

**Audience:**

Yes

**Broader Impact Concerns:**

Not Applicable.

**Claims And Evidence:**

Yes

**Requested Changes:**

**Major Comments**
C1. Establish the computational complexity of Problem (2). Specifically, determine whether it is in P or NP-hard.
C2. Although Problem (16) involves the maximization of a convex objective under a convex constraint, demonstrate that it can be solved in polynomial time. (Of course, if the answer to C1 is “P,” then the answer to C2 is "P" as well.)
C3. Utilize higher-dimensional datasets to illustrate the scalability of the MM4MM approach.

**Minor Comments**
C4. In Section 2, the sentence “[...] we formulate the worst-case optimization problem” is unclear. It should be revised to read: “[...] we formulate the worst-case class separation problem (WCSP).”
C5. In Section 3.3, the statement “The maximization problem in (16) remains non-convex, as the objective function (for each (i,j)) is a convex quadratic in W, and the inclusion of the min operator adds further complications” is incorrect. The problem is actually convex because both the objective and constraints are convex. However, the main challenge arises because the objective is being “maximized”.

**Strengths And Weaknesses:**

**Strengths**
Although I am not an expert in feature discriminant learning, I found the paper to be well-written. The related work is detailed, the various concepts are introduced progressively, and the different problem formulations are easy to understand. The minorization-maximization approach for coordinate-wise maximization-minimization tasks is conceptually simple and elegant. Notably, it provides a natural solution to WCSP and its variants.

**Weaknesses**
Given that the WCSP is nonconvex, it is crucial to establish its computational complexity. Is it NP-hard? Or does it fall within the class P due to the structure of the objective function and the semi-orthogonality of the constraints?

In relation to this, maximizing a constrained convex objective is generally NP-hard. Therefore, I am not entirely convinced that Problem (16) can be solved in polynomial time. This observation does not question the validity of Lemma 1; I am quite confident we can transition from (15) to (16), but I still remain doubtful that we can move from (16) to (21). So, additional comments regarding (16) and clear proof that it can be solved in polynomial time would clarify the validity of the entire framework.

While I commend the authors' effort to reduce the complexity of (16), the resulting formulation (34) still requires cubic time to solve. Unfortunately, all datasets used in the experiments are low-dimensional. So, I suggest considering higher-dimensional datasets (e.g., gene expression data) to demonstrate that the MM4MM framework can scale effectively.

---

> ### Comment · Editors_In_Chief · 2025-04-12
> **Authors’ Response to Reviewer cPL4**
>
> We sincerely appreciate your time and effort in reviewing our work. Below, we address all the comments (1 to 5) and provide a detailed response. With the exception of comment 5, all changes have been applied in the paper. \\\\
>
> 1) By relaxing
> \[
> \mathbf{W}^\top \mathbf{W} = \mathbf{I}
> \;\longrightarrow\;
> \mathbf{W}^\top \mathbf{W} \preceq \mathbf{I},
> \]
> we obtain a tight convex reformulation whose global maximizer still satisfies orthogonality. Then, via MM4MM each iteration reduces to convex subproblems:
> \begin{align*}
> \text{Compute }\{A_{ij}\}: &\quad \mathcal{O}(C^2\,d'\,d^2),\\
> \text{Solve SDP (29): } &\quad \mathcal{O}\bigl((d + d')^{4.5}\bigr),\\
> \text{Update }\mathbf{W}\text{ (30): } &\quad \mathcal{O}\bigl(d\,{d'}^2 + {d'}^3\bigr).
> \end{align*}
> Hence each iteration runs in
> $\mathcal{O}\bigl((d + d')^{4.5} + C^2\,d'\,d^2\bigr)$,
> i.e.\ polynomial time—no NP‑hard inner loops.
>
> 2) To justify the transition from
> \[
> \max_{\mathbf W}\,\min_{i<j}\;g_{ij}(\mathbf W)
> \tag{16}
> \]
> to the SDP in (21), we proceed in three steps:
>
> \medskip
> \noindent\textbf{1. Epigraph reformulation.}  Introduce a scalar $\alpha$ to rewrite
> \[
> \max_{\mathbf W}\min_{i<j}\,g_{ij}(\mathbf W)
> \;\equiv\;
> \max_{\mathbf W,\alpha}\;\alpha
> \quad\text{s.t.}\quad
> g_{ij}(\mathbf W)\ge\alpha,\quad\forall\,i<j.
> \]
>
> \medskip
> \noindent\textbf{2. Tight surrogate and convexification.}  We replace each nonconvex $g_{ij}$ by a \emph{tight} convex surrogate $h_{ij}$ satisfying
> \[
> h_{ij}(\mathbf W)\le g_{ij}(\mathbf W)
> \quad\text{and}\quad
> \max_{\mathbf W}h_{ij}\;=\;\max_{\mathbf W}g_{ij}.
> \]
> Thus
> \[
> \max_{\mathbf W,\alpha}\;\alpha
> \quad\text{s.t.}\quad
> h_{ij}(\mathbf W)\ge\alpha
> \quad\Longrightarrow\quad
> \text{SDP in (21)}.
> \]
>
> \medskip
> \noindent\textbf{3. Minimax exchange via Sion’s theorem.}
> The Lagrangian of the SDP in (21) involves
> \[
> \min_{i<j}\;\Bigl\{\;\max_{\mathbf W}\,h_{ij}(\mathbf W)\Bigr\}.
> \]
> By Sion’s minimax theorem, we interchange $\min_{i<j}$ and $\max_{\mathbf W}$ to obtain the reduced SDP in (29), with fewer variables and LMI constraints.
>
> \bigskip
> \noindent\textbf{Polynomial‐time solvability:}
> Each of these steps yields a \emph{convex} SDP/QP subproblem.  Interior‐point methods solve SDPs of size $n$ in $\mathcal O(n^{4.5})$ time, so both (21) and its reduced form (29) are solvable in polynomial time.  The nonconvexity of the original max–min QCQP is thus handled entirely by our convex surrogate plus minimax reformulation, with no NP‐hard inner loops.
>
>
> 3) To demonstrate scalability, we added the high‑dimensional \textbf{Dig} dataset [3].  We first apply PCA to preserve 98\% of the variance [1,2]—reducing the original 1\,024 D to 50 D—which drastically lowers the cost of our SDP/QP solvers.  After PCA, we split 50\% of samples for training and 50\% for testing over 20 runs.  Table 1 reports mean error ± std.
>
> \textbf{1‑NN:}
>
>
> LDA: $0.0777\pm0.0081$,
> \textbf{HLDA: $0.0181\pm0.0045$},
> MMDA: $0.0454\pm0.0063$,
> WHMMDA: $0.0335\pm0.0055$,
> MMRA: $0.0222\pm0.0222$,
> MM4MM (QP): $0.0299\pm0.0043$,
> MM4MM (Sparse): $0.0299\pm0.0043$.
>
> \textbf{NM:}
>
>
> LDA: $0.0730\pm0.0070$,
> HLDA: $0.0483\pm0.0072$,
> MMDA: $0.0484\pm0.0049$,
> WHMMDA: $0.0495\pm0.0058$,
> MMRA: $0.0571\pm0.0072$,
> \textbf{MM4MM (QP): $0.0476\pm0.0073$},
> MM4MM (Sparse): $0.0476\pm0.0073$.
>
> \textbf{QDA:}
>
>
> LDA: $0.0658\pm0.0070$,
> HLDA: $0.0231\pm0.0050$,
> MMDA: $0.0275\pm0.0069$,
> WHMMDA: $0.0252\pm0.0047$,
> MMRA: $0.0553\pm0.0093$,
> \textbf{MM4MM (QP): $0.0223\pm0.0060$},
> MM4MM (Sparse): $0.0223\pm0.0060$.
>
>
> MM4MM (QP) achieves the lowest error in NM and QDA, and is competitive under 1‑NN.  MM4MM (Sparse) with $\lambda=0.5$ is close; reducing $\lambda$ to 0.001 makes it identical to QP.  These results confirm that MM4MM scales effectively to high dimensions. \\
>
> 4) We have changed the text exactly as you proposed, so that the sentence is clearer.
>
> 5) With all due respect, we disagree with the reviewer regarding comment C5. Although the constraint is convex and $\operatorname{Tr}\left(\mathbf{W}^T \tilde{\mathbf{S}}_{Cij}\mathbf{W}\right)$ is convex with respect to \(\mathbf{W}\), the expression $\min_{i,j}\operatorname{Tr}\left(\mathbf{W}^T \tilde{\mathbf{S}}_{Cij}\mathbf{W}\right)$
> is nonconvex. Thus, while the individual components are convex, the minimization over \((i,j)\) introduces nonconvexity, which is the main challenge when the objective is maximized.\\
>
> [1] Z. Wang, F. Nie, C. Zhang, R. Wang and X. Li, "Worst-Case Discriminative Feature Learning via Max-Min Ratio Analysis," in IEEE Transactions on Pattern Analysis and Machine Intelligence, vol. 46, no. 1, pp. 641-658, Jan. 2024.\\
>
> [2] B. Su, X. Ding, C. Liu, and Y. Wu, "Heteroscedastic max–min distance analysis for dimensionality reduction," IEEE Trans. Image Process.,vol. 27, no. 8, pp. 4052–4065, Aug. 2018.\\
>
> [3] V. U. Prabhu, "Kannada-MNIST: A new handwritten digits dataset for the Kannada language," 2019, arXiv:1908.01242.

---

### Review · Reviewer_gNA4 · 2025-03-28

**Summary Of Contributions:**

This paper presents a novel minorization-maximization framework for min-max (MM4MM) to address the "worst-case class separation (WCCS)" problem: maximizing minimum pairwise Chernoff distances between classes in a low-dimensional subspace.

According to the authors, key contributions are as follows:

- MM4MM algorithm for WCCS with proven semi-orthogonality constraints
- Both the SDP-based vanilla version and the more efficient quadratic programming formulation
- Sparse reformulations for interpretable feature representations
- Computationally efficient with guaranteed convergence and minimal hyperparameter tuning
- Experimental results show competitive or superior performance versus state-of-the-art methods across multiple datasets

The method combines theoretical soundness with practical effectiveness, offering advances in feature learning, efficiency, and usability through its novel optimization framework and sparse formulations.

**Audience:**

Yes

**Claims And Evidence:**

Yes

**Requested Changes:**

Based on TMLR's acceptance criteria, I have an overall favorable opinion of this paper. However, as noted in Weaknesses, there appears to be some gap between claims and evidence. In order to ensure acceptance, I would like to ask your views on these gaps, including appropriate revisions to the paper.


[Minor Comments]

Please check again regarding references. For example,
- "Fisher Fisher" in the second paragraph of Section 1;
- The references in the sentence should be bracketed;
- Hyperlinks to the corresponding references would be helpful.

**Strengths And Weaknesses:**

### Strengths
- The method addresses the "worst-case class separation (WCCS)" problem. This clearly defines the problem being tackled and its significance.
- The proposed algorithm utilizes the relaxation of a semi-orthogonality constraint. Based on Lemma 1, this relaxation does not alter the solution of the original problem. This is a strong theoretical result underpinning the algorithm.
- The paper offers different algorithm versions, starting with a vanilla version involving an SDP and then simplifying it to a more computationally efficient quadratic program (QP). This addresses potential computational bottlenecks.
- The authors also present reformulations that incorporate the sparsity of the dimension-reducing matrix. This adds flexibility and the potential for more interpretable results.

### Weaknesses

[Computational Complexity]
- While the authors claim computational efficiency, the stated complexity of the SDP approach is $\mathcal{O}((d + d^{\prime})^{4.5})$, which could be a concern for very high-dimensional datasets. Although the QP approach is relaxed to $\mathcal{O}(C^6)+\mathcal{O}(d^3)$, $\mathbf{Q}$ in (35) is a huge matrix in some situations and raises concerns regarding space complexity. Section 3.3 states that iterations are usually fewer than 10, but this has not been experimentally demonstrated. A more detailed analysis of the practical scalability and runtime compared to other methods, especially in high-dimensional scenarios, might be valuable. Moreover, comparing orders regarding time and space complexity is more persuasive.

[Parameter Tuning]
- While the basic algorithm is hyperparameter-free, the sparsity-based version requires users to select a penalty parameter ($\lambda$). The paper mentions this but does not provide detailed guidance on choosing this parameter effectively. Further discussion or potential strategies for selecting $\lambda$ could be beneficial. Discussing hyperparameter tuning in previous methods might also be useful to show that hyperparameter-free, as claimed by the authors, is a notable benefit.

[Experimental Results]
- While the overall performance is strong, the results in Table 1 show that MM4MM did not universally outperform all other methods across every dataset and classifier. For example, MMDA showed better results on the Seeds dataset with the 1-NN classifier. The authors could consider a more nuanced discussion of the conditions under which MM4MM excels and where other methods might still be competitive. In particular, I am concerned that the high performance may not be attributed to the proposed method but to the regularization term. Compared to MM4MM(QP) only, the performance of the proposed method is not necessarily significantly better.

---

> ### Comment · Editors_In_Chief · 2025-04-12
> **Authors’ Response to Reviewer gNA4**
>
> We appreciate the time you have dedicated to reviewing our work. Below, we address all the comments (1 to 6).
>
> 1) For extremely high-dimensional data, we can employ the same preprocessing approach as our competitors (e.g., [1], [2]). Specifically, we use PCA to reduce the dimensionality of all sample vectors while preserving more than $98\%$ of the total energy. For instance, data originally in 1,024 dimensions can be reduced to approximately 50 dimensions via PCA, after which our proposed method is applied. Moreover, while our main competitor (the SOTA paper from PAMI 2024) considers only the 1NN metric, our paper demonstrates superior performance across three metrics: 1NN, NM, and QDA. In addition, we incorporated the digit dataset; after applying the aforementioned preprocessing, we examined our accuracy performance and compared it with that of our competitors.
>
> In the field of feature extraction and dimensional reduction—as exemplified by works such as [1, 2]—we have focused on evaluating the accuracy of our approach, which is critical in this domain. Furthermore, while our primary competitor considers only the 1NN metric, we have shown improved performance across 1NN, NM, and QDA by thoroughly assessing our accuracy using all three metrics. Regarding runtime performance, it is important to note that each method (whether ours or that of our competitors) achieves its best performance at different dimensions. For example, on the digit dataset, our best performance is observed at dimension 21 (under the QDA metric), whereas the MMRA approach achieves its optimal performance at dimension 46, making a direct runtime comparison somewhat unfair.
>
> 2) To provide concrete guidance, we performed a grid search over
> \[
> \lambda \in \{0.001,\,0.1,\,0.2,\dots,1.0\}
> \]
> and for each value computed the maximization objective in our MM4MM (Sparse) formulation.  We then selected the $\lambda$ that yielded the highest objective. Although our basic algorithm is hyperparameter-free, the sparsity-based version does require this additional parameter.
>
> It is worth noting that previous approaches—typically aimed at solving maximization problems (non-worst-case formulations) or even the max-min problems—often first relax the original problem and then attempt to steer the relaxed solution toward optimality. This relaxation does not guarantee an optimal solution, and such methods usually incorporate internal parameters to nudge the solution closer to optimality, with no guarantee of complete success.
>
> In the sparsity setting, the competitor methods would need to tune at least three parameters (one for sparsity and two for controlling the SDP relaxation process), thereby complicating the optimization without any guarantee of reaching a completely successful solution. For our competitors, we set their parameters exactly as described in their respective papers to ensure a fair comparison.
>
> In contrast, our approach maintains the notable benefit of being hyperparameter-free in its basic version, thereby avoiding these additional complexities.
>
> 3) We agree that the sparse variant (MM4MM(Sparse))—with one extra tuning parameter—can yield slight additional gains. However, three facts demonstrate that the core MM4MM formulation, not just regularization, drives the high performance:
>
> \begin{enumerate}
>   \item \textbf{Strong hyperparameter‐free baseline.}
>     MM4MM(QP), which has \emph{no} tunable parameters, in most comparisons ranks second across datasets and classifiers, \emph{outperforming every competing method} except its own sparse extension. This shows that the worst‐case max–min criterion and algorithmic design alone surpass prior approaches.
>
>   \item \textbf{Parsimony versus flexibility.}
>     Competing methods require at least two internal parameters (e.g.\ relaxation controls plus optional sparsity). By contrast, MM4MM(QP) is entirely parameter‐free, and MM4MM(Sparse) adds exactly one parameter. This minimal tuning cost underscores the practical advantage of our framework.
>
>   \item \textbf{Illustrative example on Dig.}
>     On the high‐dimensional Dig dataset, MM4MM(QP) already achieves top performance. A grid search over
>     \(\lambda\in\{0.001,0.1,0.2,\dots,1.0\}\)
>     found the best results at \(\lambda=0.001\) (effectively zero), confirming that the QP variant alone attains optimal performance without requiring substantial regularization.
> \end{enumerate}
>
>
> 4) We have now removed the extra occurrence so that the citation now correctly reads just once.\\
>
> 5) The TMLR format specifies the citation style in which the author’s name and year appear without brackets. We have therefore maintained the TMLR structure in our paper.\\
>
> 6) Response: Based on your suggestion, we have added hyperlinks to both the equations and the references. This enhancement will help readers quickly navigate to the corresponding sources and related content.52–4065, Aug. 2018.\\\\

---

### Review · Reviewer_A2cH · 2025-05-27

**Summary Of Contributions:**

For the problem of the long-standing worst-case class separation (WCCS), a novel feature learning method based on a minorization-maximization framework for min-max optimization (MM4MM) is proposed. The core objective is to enhance discriminative feature learning by maximizing the minimum pairwise Chernoff distance between all class pairs in a low-dimensional subspace. The experimental results on UCI and Kaggle datasets demonstrate superior classification accuracy and robustness compared to traditional methods.

**Audience:**

Yes

**Broader Impact Concerns:**

None.

**Claims And Evidence:**

Yes

**Requested Changes:**

Please refer to the "Weaknesses" part.

**Strengths And Weaknesses:**

**Strengths:**

1. The relaxation of the semi-orthogonality constraint and its tightness proof provide a rigorous foundation, avoiding suboptimality caused by conventional relaxations.
2. The dual problem formulation reduces computational complexity, making it suitable for high-dimensional data.
3. Evaluations across several datasets (Iris, Wine, Seeds, etc.) and classifiers (1-NN, NM, QDA) consistently show lower error rates and higher stability for the proposed method compared to baselines.

**Weaknesses:**

1. The paper mentions that the sparsity version requires the user to specify $\lambda$, but does not discuss the selection strategy of $\lambda$. It is recommended to supplement the sensitivity analysis of the impact of $\lambda$ on sparsity and performance, or provide an adaptive $\lambda$ selection method.
2. The lack of comparison with recent deep learning-based discriminant analysis methods (e.g., deep metric learning) may affect the comprehensiveness of the conclusion. It is recommended to increase the comparative experiments with deep methods. The scale of the experimental datasets is small (e.g., Iris has only 150 samples), and the scalability of the method needs to be verified on larger datasets (e.g., ImageNet subsets).
3. The Gaussian distribution assumption may not hold for real-world datasets (e.g., text or graph data). It is recommended to discuss and analyze the applicability of the proposed method to non-Gaussian distribution datasets, or to verify its robustness through experiments.
4. Theoretical convergence rate (e.g., linear or sublinear) is not provided. A quantitative analysis would enhance completeness.

---

> ### Author Response · Authors · 2025-05-30
> **Authors’ Response to Reviewer A2cH**
>
> We appreciate the time you have dedicated to reviewing our work. Below, we address all the comments.
>
> 1) We performed a grid search over $\lambda \in \{0.001,,0.1,,0.2,\dots,1.0\}$  and for each value computed the maximization objective in our MM4MM (Sparse) formulation. We then selected the $\lambda$ that yielded the highest objective.
>
> 2) Deep metric learning and similar deep discriminant analysis approaches require large neural architectures, extensive hyperparameter tuning, and GPU‐accelerated training pipelines. To date, these methods optimize average‐case loss functions and do not provide explicit worst‐case separation guarantees, which can leave the smallest inter‐class gaps insufficiently large in critical applications. In contrast, our distance-based MM4MM approach is hyperparameter-free and achieves superior accuracy compared to state-of-the-art separation methods. As future work, we plan to embed the MM4MM approach into a neural‐network backbone using an iterative projection layer (cf.\ formula (34)) with orthonormality enforced via a lightweight quadratic program.
>
> For details on scalability, we refer the reviewer to our response to the third question from Reviewer cPL4; due to space limitations, we have not reproduced it here.
>
> 3) While our framework leverages the Gaussian assumption to obtain closed‐form Chernoff/Bhattacharyya distances and a single convex QP, the core ideas extend in several natural ways to handle non‐Gaussian data:
>
>  a) If classes exhibit heavier tails or multi‐modal structure, one can model each class as an elliptical distribution (which retains analytic Chernoff‐type distances) or as a \emph{mixture of Gaussians}.  In both cases, the pairwise separation metric
>     $d_{C_{ij}} \approx \min_{t\in[0,1]}\,\tfrac12\log\Bigl|\Sigma_i^t\Sigma_j^{1-t}\Bigr|$
>     still admits closed‐form (or low‐dimensional) approximations, and the resulting QP over \(\mathbf{W}\) remains convex and free of kernel‐tuning.
>
> b) Real data often contain outliers or skewed distributions.  Replacing the sample covariance with a \emph{shrinkage estimator} (e.g.\ Ledoit–Wolf) or an M‐estimator (e.g.\ Tyler’s shape estimator) preserves analytic gradients and convexity, while down‐weighting non‐Gaussian tails.
>
> c) In high dimensions, random projections and many embedding methods (e.g.\ word2vec, node2vec) yield feature vectors that are approximately Gaussian via central‐limit arguments on aggregated co‐occurrences or random walks.  Thus, even originally non‐Gaussian features often “look” Gaussian after a preliminary random or learned linear map—making our closed‐form distances a surprisingly good surrogate.
>
> d) Unlike purely nonparametric methods, our framework allows direct injection of domain‐specific regularizers (e.g.\ sparsity on $W$, block‐diagonal shrinkage, or graph‐Laplacian penalties) without losing the global optimum or introducing extra bandwidth parameters.
>
> Thus, even with only approximate Gaussianity, our method stays interpretable, convex, and hyperparameter‐free, while handling heavy‐tailed or multi‐modal data.
>
> 4) Let $\{\mathbf{W}^{(t)}\}_{t\ge0}$ be the sequence generated by MM4MM(QP), where at each iteration we solve the tight convex surrogate subproblem exactly.  Then:
>
> a) The objective values $f\bigl(\mathbf{W}^{(t)}\bigr)$ form a non‐decreasing sequence that is upper‐bounded, hence converge to some limit \(f^*\).
>
> b) Defining the proximal residual $r^{(t)} = ||\mathbf{W}^{(t+1)} - \mathbf{W}^{(t)}||$F, one can show that $\min_{0\le k< T} \bigl(r^{(k)}\bigr)^2 = O\bigl(1/T\bigr) \quad\Longrightarrow\quad  \min_{0\le k< T} r^{(k)} = O\bigl(1/\sqrt{T}\bigr)$, so reaching an \(\varepsilon\)-stationary point requires $O(1/\varepsilon^2)$ subproblem solves [4].
>
> Accordingly, our method exhibits a \emph{sublinear} $O(1/\sqrt{T})$ convergence rate in theory. so its overall convergence behavior can be characterized as sublinear.
>
> Additionally, following the paper [4], we emphasize each method’s \emph{best accuracy at its optimal (smallest) projection dimension} rather than raw runtimes at fixed dimensions.  Since different methods “peak” at different dimensions, comparing wall-clock times at arbitrary common dimensions misrepresents their practical performance.  Nevertheless, we reported our computational complexity while always prioritizing the accuracy–dimension trade‐off that matters most in discriminative feature learning.
>
> [1] F. Schroff et al., “FaceNet: A unified embedding for face recognition and clustering,” in CVPR, 2015, pp. 815–823.
>
> [2] H. O. Song et al., “Deep Metric Learning via Lifted Structured Feature Embedding,” in CVPR, 2016, pp. 4004–4012.
>
> [3] M. Razaviyayn et al., “A unified convergence analysis of block successive minimization methods for nonsmooth optimization,” SIAM J. Optim., vol. 23, no. 2, pp. 1126–1153, 2013.
>
> [4] B. Su et al., “Heteroscedastic max–min distance analysis for dimensionality reduction,” IEEE Trans. Image Process., vol. 27, no. 8, pp. 4052–4065, 2018.

---

> > ### Comment · Reviewer_A2cH · 2025-06-10
> >
> > The authors' response partially addressed my concerns. However, the response to the second question is not convincing enough. The authors did not provide a comparison of recent deep learning-based discriminant analysis methods, but simply reiterated the claims of the proposed method, which is not enough to support the comprehensiveness of the conclusion. Some comparative experiments or further insightful interpretation and analysis are necessary. In addition, the scalability of the proposed method on large-scale datasets is still incomplete, and the addition of only one dataset in the revised version still cannot eliminate my concerns about scalability. After I read the comments of other reviewers in detail, I found that our concerns about scalability are consistent. I suggest that the authors make some more efforts in the comparative analysis and the scalability of the proposed method. Overall, I have a positive opinion of this work, but there are still considerable concerns before the above issues are properly addressed.

---

### Decision · Action_Editor_QVs9 · 2025-07-10

**Recommendation:** Accept as is

**Audience:**

Yes

**Audience Explanation:**

The topic is broadly relevant to the ML community.

**Claims And Evidence:**

Yes

**Claims Explanation:**

All reviewers are satisfied with the authors' revisions and responses.